# The Role of Oxytocin in Early-Life-Stress-Related Neuropsychiatric Disorders

**DOI:** 10.3390/ijms241310430

**Published:** 2023-06-21

**Authors:** Yue Jin, Da Song, Yan Yan, Zhenzhen Quan, Hong Qing

**Affiliations:** Key Laboratory of Molecular Medicine and Biotherapy, School of Life Science, Beijing Institute of Technology, Beijing 100081, China; 13865415969@163.com (Y.J.); songda18810276950@163.com (D.S.); 15128470659@163.com (Y.Y.); qzzbit2015@bit.edu.cn (Z.Q.)

**Keywords:** early-life stress, oxytocin, neural circuit, neuropsychiatric disorders, social behavior

## Abstract

Early-life stress during critical periods of brain development can have long-term effects on physical and mental health. Oxytocin is a critical social regulator and anti-inflammatory hormone that modulates stress-related functions and social behaviors and alleviates diseases. Oxytocin-related neural systems show high plasticity in early postpartum and adolescent periods. Early-life stress can influence the oxytocin system long term by altering the expression and signaling of oxytocin receptors. Deficits in social behavior, emotional control, and stress responses may result, thus increasing the risk of anxiety, depression, and other stress-related neuropsychiatric diseases. Oxytocin is regarded as an important target for the treatment of stress-related neuropsychiatric disorders. Here, we describe the history of oxytocin and its role in neural circuits and related behaviors. We then review abnormalities in the oxytocin system in early-life stress and the functions of oxytocin in treating stress-related neuropsychiatric disorders.

## 1. Introduction

Stress refers to an organism’s insufficient physiological response to any mental, emotional, or physical pressure, whether real or imagined [1,2]. While stress can positively impact behavior and brain health, chronic stress can also have substantial and persistent negative effects. Various forms of stress imposed at different life stages can affect individuals, such as early-life stress (ELS) and adult stress. ELS encompasses adverse experiences of the neonatal period, early and late childhood, and adolescence (e.g., abuse, neglect, loss of parental care, hunger, extreme poverty, and family/community/school violence). ELS also includes adverse fetal exposure, such as maternal malnutrition, stressful maternal living circumstances, increased maternal anxiety, and extreme adverse experiences [3,4]. ELS is conceptually broad, but here we focus on stressful early-life experiences and will refer to them as ELS [5].

Early-life stress can create mental health issues; people who experienced childhood abuse are more prone to develop depression, anxiety, post-traumatic stress disorder (PTSD), substance use disorders, and aggressive behavior as adults [6,7]. Similar to humans, animals exposed to ELS display behavioral abnormalities and are prone to depression and anxiety in adulthood. Therefore, it is essential to understand the mechanisms by which ELS contributes to a variety of neuropsychiatric diseases in order to improve human health. In early life, the developing brain is extremely plastic, and neural circuit development during this period is influenced by life experiences, particularly negative stimuli [8]. The stress response comprises a series of neural events in the hypothalamus–pituitary–adrenal (HPA) axis that trigger a neuroendocrine cascade when activated. The paraventricular nucleus releases corticotropin-releasing factor, which promotes the secretion of adrenocorticotropic hormone from the anterior pituitary, which, in turn, stimulates the production and release of glucocorticoids from the adrenal cortex. Along with this “classical” neuroendocrine response to stress, the posterior pituitary secretes oxytocin from the periphery [9,10]. The oxytocin system develops in utero and early life, during which stress exposure cause dysregulation of the oxytocin system and associated functions (mother–infant attachment, social bonding, and responses to stress, anxiety, and depression) [11,12,13]. These deficits lead to abnormal social behaviors and result in neuropsychiatric disorders.

Finding neural circuits underlying stress-induced oxytocin malfunction may inform the process by which ELS develops. Glial cells, which actively regulate synaptic development, pruning, neurovascular connection, and phagocytosis, are essential for the construction of neural circuits during this time [14]. Our review mainly discusses the oxytocin system, which has several social functions that, when regulated, lead to neuropsychiatric disorders. Oxytocin administration has been attempted for the treatment of these disorders.

## 2. The Properties of Oxytocin

Oxytocin is a neuropeptide hormone primarily synthesized in the brain by the parvocellular neurons (parvOT) of the paraventricular nucleus (PVN) and the magnocellular neurons (magnOT) of both the PVN and supraoptic nucleus (SON) [15,16,17,18]. The magnOT neurons primarily innervate the forebrain and release oxytocin through the posterior pituitary gland into the bloodstream to supply the body. The parvOT neurons release less oxytocin, mainly to brainstem nuclei, the spinal cord and amygdala, stria bed nucleus, nucleus accumbens (NAc), magnOT neurons of the SON, and other regions [19,20,21]. The oxytocin gene encodes the structural precursor to oxytocin and is expressed in the mammalian hypothalamus; oxytocin is stored in large, dense-core vesicles [22,23]. The axonal projections of PVN and SON pass through the median eminence and innervate the posterior pituitary lobe, thereby releasing OXT into circulation or through volume transfer to nerve tissue to regulate physiology [24]. Oxytocin is also released to surrounding brain regions through synaptic structures [25,26]. Oxytocin levels are regulated by hydrolysis by oxytocinase (cystinyl aminopeptidase) [27], and the local dynamics of oxytocin concentrations result from oxytocin release, local enzymatic clearance by peptidases, and/or diffusion of oxytocin via bulk flow [28].

Oxytocin works through its receptor, OXTR [29,30], which belongs to the GPCR family and can activate several different G-protein-induced signaling cascades [31,32]. OXTR activation has been associated with two separate intracellular signaling cascades with dependence on either: Gαi/o or Gαq [28,32]. OXTR-expressing neurons have different functionality in different brain regions: social reward in the ventral tegmental area (VTA) [33], social recognition in the anterior olfactory nucleus (AON) [34], and social memory in the hippocampal CA2 region [35].

OXTRs are distributed in neurons, astrocytes, and microglia [32,36]. Neuronal oxytocin signaling is primarily influenced by the amount of locally released oxytocin, OXTR affinity, density, local enzymatic cleavage, and the resulting concentration of oxytocin in the extracellular fluid [28]. In addition, the formation of homodimers or heterodimers of OXTRs with other receptors may affect the affinity and downstream signal transduction of OXTR [28]. OXTR cell-surface expression and availability for binding affect how the body reacts to oxytocin. For example, OXTRs can form heterodimers with vasopressin (AVP) receptors V1a and V2. Oxytocin generally has the highest affinity for OXTRs but also easily binds to AVPR1A [37,38]. OXTR signaling in astrocytes can change neuronal excitability and the output of the amygdala neural network and can also increase local astrocyte network activity [39].

Studies of OXTR expression in microglia have yielded conflicting results [36]; most studies thus far have been in vitro [36,40,41,42,43]. OXTR mRNA levels in microglia were reported to decrease when P5 primary microglia were stimulated with lipopolysaccharide in vitro [43], while mRNA levels increased after P1–P2 primary microglia were stimulated [40]. Macrophage OXTR expression in the inflammatory state was unaltered in 3-month-old C57BL/6 mice, so the developmental staging of in vitro test subjects may influence oxytocin responses to neuroinflammatory conditions [36]. According to RNA-Seq databases of in vivo microglia single-cell expression, there is limited evidence of microglia *OXTR* expression at any life stage [36,44,45,46]. Thus, the anti-inflammatory effects of oxytocin treatment may result indirectly from the effects on astrocytes, at least in the early stages of life [36].

Oxytocin treatment reduces inflammation and the severity of various diseases. OXTRs have been found on immune cells, including neutrophils, macrophages, and lymphocytes [47]. During inflammation, nuclear factor kappa-light-chain-enhancer (NF-κB) mediates increased OXTR expression in macrophages [48]. Oxytocin can inhibit the macrophage transition to active inflammatory cells by promoting the expression of β-arrestin 2 [47] and peroxisome proliferator-activated receptor gamma [49,50].

## 3. Oxytocin-Involved Social Behavior

Oxytocin regulates social behaviors related to altruism, parent–child bonding, reward, competition, and aggression. We consider altruistic, rewarding, and cooperative behaviors as positive behaviors and competitive and aggressive behaviors as negative. Oxytocin also plays an important role in appetite, analgesia, fear, and anxiety.

Studies show that activation of the agouti-associated protein can inhibit OXT+ neurons in the hypothalamic PVN, thus boosting appetite [51]. By inhibiting presynaptic long-term potentiation, oxytocin in the anterior cingulate cortex (ACC) reduces neuropathic pain and emotional anxiety [52]. Activation of the PVN–ACC circuit alleviates pain and anxiety-like behavior in a mouse model of neurological injury, and these effects are blocked by OXTR antagonism [52]. Recent studies suggest that oxytocinergic neurons in the PVN can exert analgesic effects by enhancing the activity of PFC neurons and regulating the transmission of pain signals [53].

### 3.1. Positive Social Interactions

Oxytocin signaling is mediated by release from afferent terminals to receptors present in various target regions that impact aspects of behavior. Oxytocin modulates a variety of positive social behaviors via OXTRs in many limbic and reward-related regions of the mammalian brain, such as the PFC, NAc, amygdala, lateral septum, and thalamus [54]. The VTA and the NAc of the limbic system of the midbrain cortex are abundant in OXTRs and are linked to dopaminergic reward motivation [33]. The VTA PVN neurons release oxytocin to dopamine-secreting neurons of the nerve bundles that stretch from the VTA to the NAc, thus strengthening the social abilities of these mice [33,55]. Oxytocin may also increase the excitatory drive of VTA dopamine neurons that project to the NAc, thus improving sociability and social pleasure and increasing 5-HT release in the NAc [33,56,57,58]. The potent monoamine releaser (±) 3,4-methylenedioxymethamphetamine (MDMA) has potent prosocial effects in humans. Direct MDMA infusion into the NAc enhances sociality in three-chamber experiments, while knocking out the 5-HT transporter gene weakens this effect, possibly because MDMA enhances 5-HT release and promotes social interaction by activating NAc OXTRs to prolong the critical developmental period of social reward learning [56,59]. The mechanisms by which neuromodulators such as 5-HT, OXT, and DA regulate key brain regions related to social behavior and reward are of great significance, and further research studying the effects of different modulators on general neural or neuronal specificity is needed.

Beyond social reward behavior, oxytocin also regulates prosocial behavior and the parent-child relationship. Countless studies have shown that oxytocin signaling regulates social interactions, especially prosocial behaviors associated with altruism [60]. MagnOT neurons have long-distance axonal projections to forebrain regions, including the PFC, AON, NAc, lateral septum, hippocampus, and medial and central amygdala, which are found only in higher vertebrates such as mammals and reptiles. These co-evolved regions are implicated in complex social and emotional behaviors [28]. In social mammals, OXT mediates prosocial behaviors such as mate preference, social approval and proximity, and parental care, and this effect is more pronounced in monogamous mating systems [61]. Social monogamy is a mating system characterized by sharing of territory among partners, mutual care, and preferential mating [62]. Neurobiological research on animal social bonds has focused on “social monogamous” species because they have strong long-term bonds, and the formation of this social bond seems to involve oxytocin [63]. The density of OXTR is higher in the NAc, mPFC, and amygdala of monogamous meadow voles than in non-monogamous montane (Microtus montanus) and meadow (Microtus pennsylvanicus) voles [62,64]. OXTR mRNA expression is present in the NAc of humans but absent from non-monogamous rhesus monkeys [62,65,66], and oxytocin is activated in the ACC and amygdala when rats engage in helping behaviors [67]. Injection of OXTR antagonists into the ACC impairs consolation behaviors for stressed partners in prairie voles [68]. Following grooming activity in chimpanzees, bond partners experience increased magnOT activity and peripheral oxytocin secretion with higher urinary oxytocin concentrations compared to non-bond partners [69,70,71,72]. An early 1979 study found that oxytocin supplementation in virgin rats can induce care-taking behavior; female rats began to build nests, lick pups, and even bring home lost pups [73]. A subsequent study found that oxytocin injection in virgin mice causes changes in neural firing patterns. Upon hearing pup cries, the irregular PVN firing patterns of virgin mice transform into a regular dam-like spike model with an increased firing frequency [74,75]. As shown in Figure 1, hypothalamic PVN oxytocin neurons projecting to the left auditory cortex regulate pup retrieval behavior in naive virgin mice [75]. Similarly, in male mice with post-mating paternity care for pups, chemical genetic activation of lateral hypothalamus projections to the PVN reduces aggression against pups in unmated male mice; this effect is diminished when oxytocin is depleted in the PVN [76]. In the medial preoptic area (MPOA) of male voles, oxytocin regulates care-taking behavior, and OXTR antagonism in this region reduces the intensity of paternal conduct [77].

Oxytocin has a crucial role in complex social behaviors such as generosity, empathy, and collaboration [78]. Wild-type zebrafish freeze when observing a conspecific in pain in an isolated tank, whereas mutant zebrafish lineages lacking OXT or OXTR or do not exhibit these fear responses, indicating that oxytocin is necessary and sufficient for social fear contagion in zebrafish [79]. Interest in the role of oxytocin in social cognition and emotional processing increased after a study found that an oxytocin nasal spray promotes trust and play in social situations [80]. Complex social behaviors are closely connected with analgesia, and the role of oxytocin in analgesia and fear is noteworthy. The lactation-induced and oxytocin-dependent lack of social dread is blocked by chemogenetic suppression of DREADD-expressing OXT+ neurons that project to the lateral septum in breastfeeding mice [81]. The freezing caused by conventional fear training is reduced by oxytocin in the central amygdala (instead of a maternal context) [21]. The lateral portion of the central amygdala (CeL) receives lateral branch axons from magnOT neurons in the PVN and accessory nuclei, and these neurons create glutamatergic synapses with OXTR-expressing GABAergic neurons (Figure 2) [21]. When OXTR-expressing CeL neurons project to the medial part of the central amygdala (CeM), the CeM neurons release GABA to improve inhibitory synaptic transmission [82].

Stressful situations and corticosterone infusion increase oxytocin and OXTR binding in the ventral hippocampus [83], and oxytocin affects inhibitory interneurons to modify the functional activity of excitatory networks in the hippocampus (Figure 2) [84]. Oxytocin can increase the firing of inhibitory PV+ hippocampal interneurons and enhance spike transmission in hippocampal pyramidal neurons through the modulation of fast-spiking interneurons [85]. High-frequency (50 Hz, blue light) stimulation of channelrhodopsin-2 OXT+ terminals in the CeL reduces the freezing response in fear-conditioned rats, possibly by activating local GABA neurons [21,86]. Selective blue-light activation of PVN parvocellular OXT+ neurons projecting to the spinal cord in rats inhibits nociception and promotes analgesia [87]. Similarly, anxiety plays an important role in complex social behaviors, and oxytocin plays striking roles in emotion regulation in anxiety and stress resistance. OXTRs within the central amygdala may form a heteroreceptor complex with dopamine D2 receptors, which can promote the cascade coupling of MAPKS and Ca^2+^-dependent calcineurin signaling, thus enhancing the anxiolytic effect of oxytocin [88]. Oxytocin-induced anxiolytic effects are blocked by dopamine D2 receptors antagonism [88], and intranasal administration of oxytocin suppresses amygdala resonance in response to social fear signals in human males and macaques [89,90,91]. This makes oxytocin a potential treatment for mental health disorders, especially those of social behavior and social cognitive interruption, such as autism spectrum disorder (ASD), social anxiety disorder (SAD), PTSD, and bipolar disorder.

Despite the growing interest in using oxytocin to treat neuropsychiatric disorders such as autism, schizophrenia, mood, and ASD, study outcomes have been inconsistent, and clarity is lacking regarding the involvement of oxytocin in neuropsychiatric function. In addition, oxytocin is cleared rapidly (half-life < 10 min) and cross-binds with AVP receptors at high doses.

### 3.2. Negative Social Interactions

Oxytocin can also produce negative behaviors such as aggression [89], and such effects are susceptible to individual and sex-specific influences. The OXT+ PVN–CeL projection helps to distinguish between positive and negative emotional states [92]. Inhibition of OXT+ projections to the CeL suppresses fear-subsiding behavior [93]. Oxytocin in the lateral septum of female mice can promote aggressive behavior in a manner that depends on environmental factors [94]. Oxytocin administration into the BNST causes unstressed mice to display social anxiety behaviors. As Trainor said, the effects of oxytocin on pro-social or anti-social behaviors vary by brain region [95].

### 3.3. Sexual Dimorphism of Oxytocinin Social Behavior

Sexually dimorphic behaviors may result from sex-specific patterns of activity in the social-behavior-related brain areas. Sexual dimorphism should be considered when investigating the oxytocin system because OXTR expression is regulated by sex hormones, though there are no evident neuroanatomical differences in the distribution of OXT and OXTR expression. There have been reports describing sex-differentiated OXT+ fibers in the MPOAi of Mongolian gerbils, where females have a higher OXT+ cell density than males [96,97,98,99], and this sex difference may result from differences in axonal transport or behavioral response speed. The OXTR+ mPFC interneurons have obvious sexual dimorphism in social behavior; activation of these neurons causes anxiety in males but promotes social behavior in females [100,101]. These neuromodulators may have sex-specific effects on social behavior [56].

## 4. Effects of ELS on the Oxytocin System and Central Nervous System

ELS is a stress-induced deficit in social behavior that is closely related to limbic abnormalities that cause chronic activation of physiologic stress responses [56]. The term “early life” is frequently used to characterize several developmental periods, such as prenatal, early postnatal (until weaning on postnatal day 21), and puberty (postnatal day 25–35 in rodents). Experience shapes neural circuits during crucial periods of development, allowing individuals to adapt their behaviors to their surroundings in a unique way [102]. During this period, the nervous system is extremely sensitive to specific environmental stimuli [103], and this sensitivity is necessary for the normal formation of neural circuits and biological learning.

As shown in Figure 1, ELS may have lasting or even permanent effects on brain function, affecting a variety of emotional, social, and cognitive behaviors and triggering behavioral abnormalities [104]. Stress-induced changes are influenced by a variety of factors, such as time, severity, and duration [105].

In humans, the effects of ELS are similar to those in children born to mothers who experience adverse living circumstances during pregnancy, children in orphanages, and children who are abused in childhood. ELS can be studied more easily by observing animal models, and there are many ELS mouse models; common models include maternal separation/maternal deprivation and limited bedding and nesting. Maternal separation is widely used to simulate ELS through repeated separations from the mother over prolonged periods. Maternal separation can induce depression-like and anxiety-like behaviors in rodents [106,107]. The ELS pups will activate the HPA axis after long-term separation from their mothers, causing corticosterone levels to rise, and these inflammation episodes may develop over time into chronic inflammation [108]. In ELS mouse models, stress exposure causes atrophy of the apical dendrites of pyramidal neurons in the mPFC of males, and the volume of the hippocampus is decreased. Similar transformations occur in humans, with cortical and hippocampal volumes changing with the duration of the stress [105]. In animal models, maternal-deprivation-induced distress impedes the integration of normal maternal sensory input and affects central nervous system function [105,109]. Maternal deprivation affects pain-related behavior in puppies, has a detrimental effect on the maturation of the sensory–spinal pain system, and may lead to hypersensitivity in early adulthood [110]. These findings demonstrate the damaging effects of ELS on brain development.

The harms to the oxytocin neuronal network are currently understudied; though ELS clearly impairs several key brain areas associated with social behavior, the connection between these brain regions and oxytocin dysfunction remains puzzling. Here we review oxytocin-associated social neural circuits that may underlie these impairments.

### 4.1. Effects of ELS on the Oxytocin System

The oxytocin system is crucial to social behavior, and its disturbance can have far-reaching implications. In prairie voles, it has been demonstrated that increased parental care causes hypomethylation of the oxytocin receptor gene [111]. Female rats raised by attentive mothers who frequently licked or groomed their paws showed significant OXTR binding [112]. During development, slow, gentle stroking can elicit pleasurable sensations and social rewards by activating C-tactile fibers [113]. This stroking also acutely activates hypothalamic OXT+ neurons and promotes OXT release, and parental care modulates hypothalamic oxytocin concentrations in rat pups [114]. Deprivation of touch and social interaction can lead to irreversible deficits in emotional, social, and cognitive behavior [115]. The oxytocin system is affected long after ELS stimuli such as prenatal and postnatal sensory deprivation (beard deprivation, dark feeding), alcohol exposure, and post-weaning social isolation have subsided [116]. In rhesus monkeys, maternal deprivation increases anxiety during isolation, suggesting that OXTRs are involved in controlling maternal deprivation-induced emotional behavior in primates [117]. A study of the oxytocin system in different maternal isolation protocols and rodent models suggests that OXTRs are involved in MD-induced abnormalities in primate emotional behavior [116]. Postnatal maternal separation reduces the number of hypothalamic OXT+ neurons and OXTR expression and oxytocin levels in some brain regions [116]. Maternal neglect in female rat offspring is associated with reduced OXTR expression in the MPOA, PVN, CeA, and lateral septum [118]. Maternal separation disorder involves the expression of OXT+ neurons, OXTR binding, and plasma OXT levels in key brain regions related to depression [119].

However, a detailed understanding of how these multiple brain regions interact to produce the physiological response to stress is still lacking. OXTR expression is downregulated under long-term isolation stress [82,120,121,122]. Early studies of oxytocin focused on changes in oxytocin levels; oxytocin levels in PVN increase under acute pressure [123]. Skin-to-skin contact, individualized care, environmental enrichment, and music exposure all benefit the development of the newborn brain [36,124,125,126,127], and all of these interventions increase oxytocin levels in infants and parents [126,128].

### 4.2. Abnormalities in Oxytocin System Altered by ELS Correlated with Glial Cells

The complex mammalian central nervous system results, in part, from the varied cell types formed during development; oxytocin may regulate cell growth, differentiation, and contact with other cells [129]. ELS can cause neuroinflammation that damages the developing brain, which then enters an overactivated state that can also be induced by bacterial, environmental chemicals, or neuronal injury or death [130]. Astrocytes are the primary cell type implicated in the neuroinflammatory response; they account for 20–40% of all glial cells, and the brain’s balance of nutrient delivery and metabolism is dependent on astrocytes. While astrocytes contain many small fibers that penetrate into the local environment and react to various stimuli, neurons have distinctive dendrites and axons that allow long-distance projections [131]. While astrocyte calcium transients can last anywhere from minutes to hours, neuronal electrical activity lasts only a few milliseconds [131]. Optogenetic photostimulation of CeA axon terminals causes the release of OXT, which creates calcium transients in adjacent astrocytes [39]. In the hippocampus, mPFC, and ACC, ELS decreases the number of astrocytes [132,133]. Microglia are also important to the neuroinflammatory response; microglia secrete the pro-inflammatory cytokines IL-1, TNF-α, and C1q to activate astrocytes [134]. Microglia are highly sensitive to the neural environment, and although there is evidence that OXT affects microglia responsiveness in neuroinflammation, the mechanism of this influence is unclear [36]. Glial cells have functions beyond support; astrocytes and microglia release neuromodulators, and oligodendrocytes produce myelin that facilitates neurotransmission and neuronal oscillations [135,136,137,138]. Oxytocin treatment improves function in these brain regions, including the amygdala [139]. Astrocytes are important for θ rhythm synchronization in the hippocampus and prefrontal cortex [140]. Following maternal separation, microglia decrease in number and become overactive [133]. Although microglia activation can become chronic and harmful to healthy tissue, the inflammatory response is initially effective in isolating and minimizing immediate toxicity [130,141]. Microglia are implicated in inflammation, and because glial cells regulate synaptic function and neural circuit development, changes in glial cells and neurons may combine to change neuronal activity, and these changes may underlie circuit-level functional changes in ELS in adulthood [140].

## 5. Potential Therapy Strategies of Oxytocin in ELS-Related Neuropsychiatric Disorders

Early-life stress can impair social behavior and contribute to a variety of stress-related diseases. Oxytocin is released into several brain regions closely associated with stress-related disorders, such as the amygdala, hypothalamus, hippocampus, and NAc. Social rewards are important for social interaction, and social reward disorder is closely related to neuropsychiatric disorders in stress-related diseases. Oxytocin is closely connected to social reward and plays an important role in stress-related neurological disorders. At present, oxytocin has been studied in numerous clinical trials of social disorders-related diseases. Dysregulation of dopaminergic signaling is associated with a variety of neuropsychiatric and neurological disorders, including ASD, Parkinson’s disease, and depression [142]. When compared to healthy control subjects, some patients with dopamine-dependent disorders (e.g., Parkinson’s disease or schizophrenia) have abnormal peripheral (plasma) and central (cerebrospinal fluid) OXT levels as well as fewer OXT+ neurons in the hypothalamus (in post-mortem tissue) [143]. Abnormalities in the OXT system may result from disease and lead to behavioral abnormalities, suggesting a role for oxytocin in these diseases.

For these reasons, oxytocin has been investigated clinically as a treatment for these stress-induced mental disorders, though oxytocin does not easily cross the blood-brain barrier. Delivery of oxytocin has been attempted via intranasal administration, though details have not been confirmed [84]. Intranasal oxytocin can reach CSF and blood circulation, though it is unclear that oxytocin can reach concentrations in the brain that would produce clinically meaningful behavioral effects; microdialysis methods are questionable and have not been performed in humans. However, a recently developed mechanism enables oxytocin transport from the periphery to the central nervous system. After intranasal, subcutaneous, or intravenous administration, oxytocin level rises in the amygdala, hypothalamus, and other regions because RAGE, a membrane-associated receptor of advanced glycation end products, binds to and transports peripheral oxytocin via endothelial cells [84,144]. Researchers recently developed a new fluorescent oxytocin sensor called MTRIAOT and a genetically encoded G-protein-coupled receptor activation-based (GRAB) OXT sensor called GRABOT1.0 that can detect oxytocin dynamics in the mouse brain [145]. Oxytocin release in the AON is induced optogenetically, and the MTRIAOT signal in the mouse AON gradually increases with laser power [145]. MTRIAOT can detect changes in extracellular oxytocin in real-time and measures endogenous oxytocin release in vivo with high sensitivity and fast dynamics. GRABOT1.0 is capable of imaging in vitro and in vivo oxytocin release with appropriate sensitivity, specificity, and spatiotemporal resolution; GRABOT1.0 has faster dynamics than MTRIAOT, making it potentially more suitable for in vivo applications [146]. Studies using GRABOT1.0 during different stages of male sexual behavior reveal that oxytocin is released from cell bodies and dendrites of OXT+ neurons in the PVN, VTA, and PFC, the target regions of OXT+ axons [146]. This new method allows the detection of subtle changes in oxytocin function that will further our understanding of the influence of oxytocin on behavior.

### 5.1. Autism Spectrum Disorder

ASD is characterized by limitations in repetitive behavioral patterns, poor social interaction, and negligible perception. Both ASD and antisocial disorder have deficits of empathy and social cognition deficits. Because OXTR is distributed primarily in social-behavior-connected brain regions, such as the olfactory bulbs, lateral septum, and piriform cortex, oxytocin may regulate social behavior in ASD [147]. Children with ASD have lower plasma oxytocin levels but higher precursor levels than healthy controls, suggesting that oxytocin processing may contribute to ASD [148]. Autism may be connected to dysfunction of the amygdala, a major component of the cortico–striatal–thalamo–limbic system and emotional circuit involved in regulating emotional stress; oxytocin treatment can reduce amygdala activity and the fear response [82]. In human clinical studies, children with ASD have lower endogenous oxytocin in their plasma and saliva, but this effect does not persist into adolescence and adulthood, suggesting that a developmental time course exists for the role of oxytocin in this neurodevelopmental disorder [149].

Clinical trials have examined the potential of oxytocin as a treatment for ASD, but the findings have been conflicting (Table 1). In one study, 32 ASD children aged 6–12 years were given intranasal oxytocin for 4 weeks, and this treatment improved the social skills of ASD children [150], while a study of 16 ASD patients aged 12–19 showed that oxytocin treatment improved emotion recognition [151]. In another study of 15 adult ASD patients, 4 h of intravenous infusion improved social information processing [152], but another study reported no appreciable social behavior change [153]. Notably, the differing dosage, timing, and duration of oxytocin administration in these studies may explain the contradictory outcomes. Oxytocin therapy for ASD is generally promising, but additional studies are needed to identify the most effective delivery methods.

Though clinical studies indicate the therapeutic potential of oxytocin for social-disorder-related diseases such as ASD, the mechanism of these effects is still unclear. Social behaviors can be studied in rodent models to measure defects related to brain function and disease.

Numerous animal studies have connected ASD and oxytocin, and clinical studies suggest a connection between ASD and inflammation; ASD patients have increased levels of pro-inflammatory cytokines in the brain (e.g., TNF-α, IFN-γ, and IL-6) [169]. In ASD animal models, microglia activation and increased peripheral and central TNF-α, IL-1β, and IL-6 have been observed [170], and the anti-inflammatory effects of oxytocin may play a role in this. Plasma oxytocin levels of male ASD patients are negatively correlated with the inflammation-related molecule IFN-γ-induced protein-16 [171,172]. In a valproic-acid-induced ASD model, oxytocin treatment significantly reduced markers of inflammation and oxidative stress and restored antioxidant enzyme activity and glutathione levels [173]. Oxytocin treatment improved autism-like symptoms in an ASD mouse model by ameliorating oxidative stress and inflammation [173,174]. Reductions in OXT+ neurons and oxytocin levels in CSF and brain have been observed in newborn ASD model mice (Magel2 mutants) and adolescent rats exposed to valproic acid [175,176]. The oxytocin metabolite OXT4-9 produces dose-dependent increases in prosocial behavior without altering anxiety levels in a mouse model of ASD [177], while increased levels of oxytocin intermediates in the PVN of Magel2-KO mice indicate defective processing of oxytocin [36,175].

A disorder of the central oxytocin system was found in Cntnap2 knock-out mice. A central hypothesis of ASD etiology is that long-term developmental disconnection causes abnormal resting-state functional connectivity, and this long-range disconnection may result from developmental events [178]. Patients with ASD have reduced functional connectivity in the cerebellum, fusiform gyrus, and occipital brain, and have lower levels of medium- and short-range functional connectivity in the posterior cingulate cortex and mPFC, indicating the distance-dependence of ASD dysfunction [179]. In the Purkinje neuron TscI ASD animal model, atypical right crus I-inferior parietal lobule structural connection was also present [179], and there are abnormalities of resting-state functional connectivity in the *Cntnap2* knock-out ASD model mice; administration of oxytocin normalizes network connectivity in these mice [180].

Many studies have examined oxytocin as a treatment for ASD, though results have been inconclusive; a large randomized trial demonstrated that oxytocin did not improve social and cognitive function in children with ASD. However, a 2022 study with ASD model mice demonstrated that oxytocin modulates the vomeronasal nucleus and alters social behavior [153,180].

### 5.2. Schizophrenia

Schizophrenia is a neurodevelopmental condition with genetic predispositions and an origin connected to stress during critical periods of development [181]; the condition has positive symptoms (delusions, hallucinations, thinking abnormalities), negative symptoms (anhedonia, sadness, social isolation, faulty thinking), and cognitive dysfunction [182]. Previous studies have identified oxytocinergic dysfunction in people with schizophrenia, and single-nucleotide polymorphisms in the *OXT* gene contribute to schizophrenia vulnerability [183]. Thus, there has been growing interest in the use of oxytocin as a treatment for schizophrenia [181]. A study of oxytocin treatment in people with schizophrenia found improvements in emotion recognition [157] and cognition [184]. The positive and negative symptoms of schizophrenia improve significantly with sustained intranasal oxytocin administration paired with antipsychotic medications, but major improvements were also realized with just a single dose of intranasal oxytocin [185].

Social cognitive dysfunction leads to exacerbated delusions, anhedonia, diminished motivation, and disengagement from social interactions, which further leads to comorbid depression [185,186,187]. Mice with knock-outs of either *OXT* [188,189,190] or *OXTR* [148,186] have deficits in social recognition, and oxytocin supplementation to the preoptic region rescues these deficits [189,191].

However, some clinical studies have produced underwhelming and variable results. Studies of endogenous central and peripheral oxytocin levels in schizophrenia patients have produced conflicting findings; some studies have found lower oxytocin levels in people with schizophrenia than in healthy people [192,193,194,195,196], while other studies report the opposite [197,198]. This variability may arise from the use of peripheral oxytocin measurements as a proxy for determining central oxytocin levels and from study-to-study variability in the dose, frequency, and duration of oxytocin administration. Overall, oxytocin has complex interactions with other functional systems, and further research is required to understand the potential of oxytocin treatment for schizophrenia [185].

### 5.3. Social Anxiety Disorders

Social anxiety disorder (SAD) is characterized by social fear, avoidance, cognitive dysfunction, and life interference, and oxytocin inhibits the amygdala’s response to fear signals, slows maladaptive cognition toward exposed tasks and reduces negative self-evaluation after social stress. In healthy subjects, intranasal administration of synthetic OXT reduces anxiety levels and broadly promotes human social behavior [199]. One meta-analysis found that a single dose of intranasal oxytocin reduced cortisol in patients with HPA axis dysregulation but did not significantly affect cortisol responses to stress stimuli in healthy individuals [200]. Many neuropsychiatric disorders, such as anxiety disorders, are characterized by dysregulation of the HPA axis and may benefit from the stress-reducing anti-stress and anxiolytic effects of intranasal oxytocin [201]. By improving amygdala–mPFC and amygdala–ACC connectivity, intranasal oxytocin alters anxiety-regulating brain regions in patients with generalized anxiety disorder [202]. However, other studies of SAD have found no positive therapeutic effect of oxytocin administration [203]; differences among study participants, test environments, and baseline anxiety levels may influence the effects of oxytocin on other physiological processes [204,205]. Though uncertainty exists regarding the therapeutic potential of intranasal oxytocin in the treatment of neuropsychiatric diseases, many preclinical and clinical studies suggest that further study may yield surprising results.

## 6. Conclusions

Early-life stress contributes to many social disorders by altering OXT and OXTR expression in adulthood. Complex social interactions and behaviors are governed by many neural circuits and neuromodulators, and oxytocin plays a crucial role in the mother–infant relationship and stress-induced neuropsychiatric disorders. The impact of oxytocin administration depends on patient gender, brain region, dosage, and experimental paradigms for ELS. Oxytocin is mainly being considered for the treatment of ASD; studies of oxytocin treatment for stress-related neuropsychiatric disorders have produced inconsistent results, but further study of this inconsistency is warranted. Oxytocin signaling may play a more limited role than previously thought in attachment behaviors that may be too essential to rely on simplistic regulation, and other regulatory pathways may compensate for defective oxytocin signaling. This may also explain the mixed results of oxytocin treatment for stress-related neuropsychiatric disorders.

## Figures and Tables

**Figure 1 ijms-24-10430-f001:**
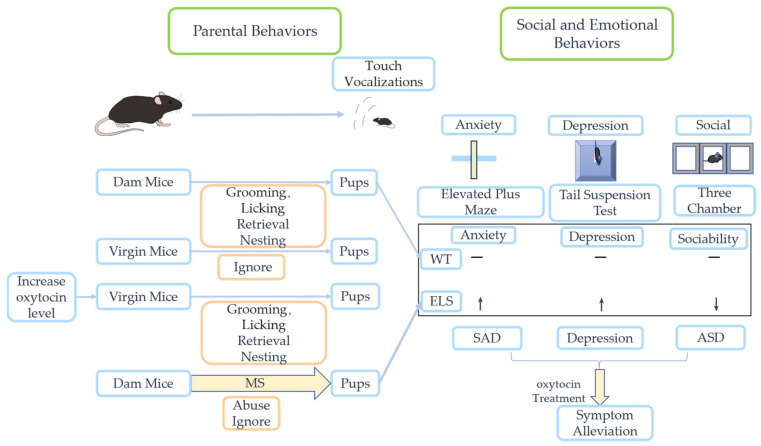
Social and emotional abnormalities in pups experiencing ELS and the role of oxytocin therapy. Generally, dam mice exhibit parenting behaviors such as nest building, pup retrieval, grooming, and licking, whereas unproductive virgin mice generally neglect pups. Virgin mice can learn parenting skills upon oxytocin administration. When mothers are briefly separated from pups daily without interruption, dam mice will neglect pups, and this establishes an ELS model for mice. ELS mice have altered levels of anxiety, depression, and socialization compared to normal mice in many experiments, and this may contribute to the increased vulnerability to stress-related neuropsychiatric disorders such as SAD, depression, and ASD in both ELS mice and humans in later life. In these disorders, treatment with oxytocin partially alleviates symptoms. ELS—early-life stress; WT—wild type; MS—maternal separation; SAD—social anxiety disorder; ASD—autism spectrum disorder.

**Figure 2 ijms-24-10430-f002:**
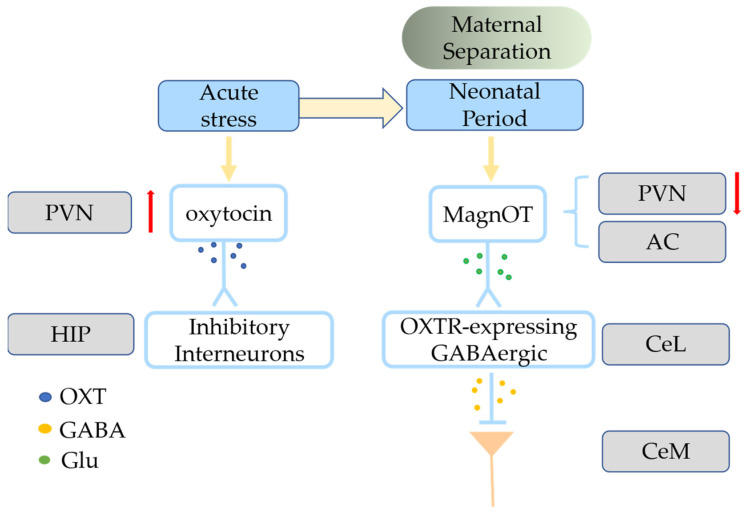
In stressful situations, disruption of oxytocin causes changes in the hippocampus and amygdala. Oxytocin levels rise in the PVN under acute stress conditions, and magnOT neurons decrease activity in the PVN of ELS mice. Oxytocin released from the PVN can modulate inhibitory interneurons in the hippocampus and magnOT in the PVN project to CeL and form glutamatergic synapses with OXTR-expressing GABAergic neurons. OXTR-expressing neurons in the CeL can project to the CeM. HIP—hippocampus; CeL—lateral portion of the central amygdala; CeM—medial part of the central amygdala; AC—accessory nuclei. Up arrow: increase; Down arrow: decrease.

**Table 1 ijms-24-10430-t001:** Therapeutic role of oxytocin in multiple stress-related neuropsychiatric disorders.

Diseases	Therapies	The Role of Oxytocin
Autism Spectrum Disorder, ASD	1. Intranasal 2. Oxytocin infusion 3. Intranasal 4. Intranasal 5. Oxytocin infusion 6. Intranasal 7. Intranasal	1. Improving the social skills of children with ASD [150] 2. Single doses of oxytocin administered to individuals with ASD improve the processing of social information [152] 3. Improved emotion recognition [151] 4. Improved social learning 5. Significant reduction in repetitive behaviors [154] 6. Improved social responsiveness of children with ASD [155] 7. No significant changes in the primary outcome measures, but after 6 weeks, results suggested improvements in measures of social cognition [156]
Schizophrenia	1. Intranasal 2. Intranasal	1. Improvement in global emotion recognition [157] 2. Reduced scores on the positive and negative symptom scale and clinical global impression-improvement scale [158]
Depression	1. Intranasal 2. Intranasal	1. Reducing depressive symptoms [159] 2. Oxytocin increased response bias for happiness and reduced negative thoughts with postpartum depression [160]
Borderline Personality Disorder (BPD)	1. Intranasal 2. Intranasal	1. Reduced vigilant attention to social threat cues [161] 2. Decreasing trust and cooperation in BPD patients [162] 3. After treatment, BPD patients showed more affiliative behavior [163]
Huntington’s Disease	1. Intranasal	1. Oxytocin normalized brain activity to disgusted faces in Huntington’s disease carriers [164]
Social Anxiety Disorder (SAD)	1. Intranasal 2. Intranasal 3. Intranasal	1. Suppressing the response to fear signals by inhibiting the amygdala [165] 2. Decreases maladaptive cognitions about performance in exposure tasks over time [166] 3. Oxytocin moderates negative self-appraisals after a social stressor for high trait, anxious male participants [167]
Post-Traumatic Stress Disorder (PTSD)	1. Intranasal	1. Reduced amygdala response to emotional faces in PTSD patients [168]

## Data Availability

Not applicable.

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
