# Peer review of "The Role of Oxytocin in Early-Life-Stress-Related Neuropsychiatric Disorders"

_ijms, 2023, doi:10.3390/ijms241310430_

Round 1
Reviewer 1 Report
In this article, the authors reviewed the associations between stress in general and in early life in particular and oxytocin. The study topic seems interesting and important. However, some changes may improve the overall quality of the manuscript.
1. The manuscript requires an overall language editing, in particular the wording should sound more scientific.
2. The abbreviations, especially ELS and OXT are not consistent within in the whole text.
3. In general the manuscript is quite long and difficult to read. A short introduction paragraph about the background between the association stress and oxytocin and the ratio and structure for this review would be helpful.
4. The title and abstract is leading to the suggestion that the review is dealing overall with early life stress. I think the paragraphs about the treatment of mental disorders with oxytocin present another topic not relevant to the initial question.
5. The introduction part should be restructured as it is complicated to read about stress in adulthood, then again in early life and associations to symptoms/diseases.
6. In my opinion the parts about biological pathways of oxytocin should be shortened and focus on the relevant relations to the stress-axis. Maybe also a figure can simplify the readability.
The manuscript requires an overall language editing, in particular the wording should sound more scientific.
Author Response
Point 1: The manuscript requires an overall language editing, in particular the wording should sound more scientific.
Response 1: We apologize for the poor language of our manuscript. We worked on the manuscript for a long time and the repeated addition and removal of sentences and sections obviously led to poor readability. We have now worked on both language and readability and have also involved native English speakers for language corrections. We really hope that the flow and language level have been substantially improved.
Point 2: The abbreviations, especially ELS and OXT are not consistent within in the whole text.
Response 2: Thanks very much for this comment. We are sorry for this mistake. We have carefully checked and corrected abbreviations throughout the text.
Point 3: In general, the manuscript is quite long and difficult to read. A short introduction paragraph about the background between the association stress and oxytocin and the ratio and structure for this review would be helpful.
Response 3: Thanks for this comment. We have adjusted the full-text sentence of the manuscript, and at the same time deleted some content of the manuscript for better understanding.
Point 4: The title and abstract is leading to the suggestion that the review is dealing overall with early life stress. I think the paragraphs about the treatment of mental disorders with oxytocin present another topic not relevant to the initial question.
Response 4: Thank you for your suggestion. Our content is mainly to explain that the oxytocin system is damaged due to early social pressure, and oxytocin is closely related to social behavior. The damage to the oxytocin system leads to neuropsychiatric disorders in the later stage, and oxytocin The effect of vitamin supplementation on the treatment of this mental disorder. It may be that the overall structure of the manuscript is not clear enough to lead to your opinion, we have worked hard to revise the content of the manuscript.
Point 5: The introduction part should be restructured as it is complicated to read about stress in adulthood, then again in early life and associations to symptoms/diseases.
Response 5: Thank you a lot for your comments. We've reorganized the introductory section and removed the confusing message about adulthood stress after careful consideration.
Point 6: In my opinion the parts about biological pathways of oxytocin should be shortened and focus on the relevant relations to the stress-axis. Maybe also a figure can simplify the readability.
Response 5: Thank you very much for your suggestion, we have added the content about stress-axis in the introduction, and also shortened the biological pathways of oxytocin.
Reviewer 2 Report
Dear authors, I have read the manuscript entitled "The role of oxytocin in early life stress-related psychiatric disorders". The review paper focuses on the physiological roles of oxytocin in the brain, as well as a number of stress-related neuropsychiatric disorders. As shown by numerous preclinical and clinical studies, oxytocin can currently be considered an important target in the treatment of these conditions.
The paper is structured according to the requirements of the journal, each section being sufficiently described as an input of information. I appreciate that the authors also gave us some conclusions. The bibliographic references are numerous, some of them being the most recent studies. The 2 figures and the presence of the table in which all the presented conditions are reproduced, make reading the manuscript easier.
However, I have a number of suggestions and questions:
1. I would start with the title: the authors defined the conditions presented as psychiatric disorders. However, I think it would be more appropriate to call them neuropsychiatric disorders. Think Huntington's disease.
2. The whole manuscript is difficult to read because of the wording. There are countless sentences in which, because of the expression, the idea presented is not clear.
3. Try not to use a term twice in the same phrase or sentence. It doesn't sound good.
4. Check figure 1. Keep the same rules in technical writing. For example, in some places you wrote in capital letters, in other places in small letters.
5. What program did the authors use to create the figures?
6. Conclusions are welcome. However, they must contain some essential ideas of the entire manuscript. I have seen bbl references. in this section. Ideas that have bibliographic references, try moving them to other sections.
7. English must be improved!
English must be improved!
Author Response
Point 1: I would start with the title: the authors defined the conditions presented as psychiatric disorders. However, I think it would be more appropriate to call them neuropsychiatric disorders. Think Huntington's disease.
Response 1: Thanks very much for this comment. We have replaced psychiatric disorders in the manuscript with neuropsychiatric disorders.
Point 2: The whole manuscript is difficult to read because of the wording. There are countless sentences in which, because of the expression, the idea presented is not clear.
Response 2: We apologize for the lack of clarity in the manuscript. Efforts have been made to reorganize and write the manuscript both syntactically and structurally.
Point 3: Try not to use a term twice in the same phrase or sentence. It doesn't sound good.
Response 3: Thank you for your suggestion, we have replaced some words with similar words.
Point 4: Check figure 1. Keep the same rules in technical writing. For example, in some places you wrote in capital letters, in other places in small letters.
Response 4: Thanks very much for this comment. We have adjusted Figure 1 to maintain the same rules when writing, and also unified the use of letter capitalization.
Point 5: What program did the authors use to create the figures?
Response 5: We used PowerPoint to create my images, and I then exported the vector graphics to include in the article.
Point 6: Conclusions are welcome. However, they must contain some essential ideas of the entire manuscript. I have seen bbl references. in this section. Ideas that have bibliographic references, try moving them to other sections.
Response 6: Thank you very much for your suggestion, we have made some changes and moves in the conclusion section.
Point 7: English must be improved!
Response 7: Thanks very much for your comments. We have asked a well-established expert, to polish our paper. Please see if the revised version met the English presentation standard.
Round 2
Reviewer 1 Report
The manuscript has improved.
Language improved; however, some sentences are still a bit hard to readability.
Reviewer 2 Report
Dear authors, I have reread the manuscript entitled "The role of oxytocin in early life stress-related psychiatric disorders". I note that you have answered all the items and that the new version of the manuscript has been considerably improved.
Just one small observation I would have:
1. In the Conclusions section, a bibliographic reference will still remain. Try to remove it, and if the phrase related to the reference does not belong to you, then formulate one of your own.
English has been improved.